# Board Games in Improving Pupils' Speaking Skills: A Systematic Review

Catherine Hui Tiing Wong and Melor Md Yunus *

Faculty of Education, Universiti Kebangsaan Malaysia, Bandar Baru Bangi 43600, Selangor, Malaysia;
p105783@ukm.edu.my
* Correspondence: melor@ukm.edu.my

**Abstract:** English is a fundamental language to learn as it is used worldwide. The teaching and learning of English has been emphasized in Malaysia as English plays a major role in global communication. However, speaking performance was recorded as poor and weak among pupils in ESL classrooms. Previous researchers explored a myriad of communicative language activities to improve speaking skill. Board games are employed as one of the most useful tools to improve speaking skills among pupils. This systematic review was conducted to examine pupils' perceptions on the use of board games in learning and speaking, as well as the usefulness of board games in improving their speaking skill. A total of 35 articles, from the period 2017–2021, were accessed through three databases: Google Scholar, ERIC and SAGE Journals. The review was conducted systematically based on the guidelines from the PRISMA statement (Preferred Reporting Items for Systematic Reviews and Meta-Analyses). Based on the articles gathered, the results showed that the qualitative research design was mostly used to collect pupils' opinions on the use of board games, while quantitative research design was mostly used to evaluate the usefulness of board games in improving speaking. Overall, the systematic review revealed that board games had several positive impacts in pupils' speaking performances such as improving speaking competency, enhancing motivation in speaking and increasing interpersonal interaction among pupils. It is suggested that future studies could focus on investigating teachers' opinions on the use of board games in teaching and speaking.

**Keywords:** board games; speaking; education; English; pupils' perceptions

## 1. Introduction

In the era of globalization, English takes its stance as the predominant language, spoken by around 400 million people across the globe. English is used as a means of verbal and written communication worldwide to bridge the gaps in economic, political and social aspects. Since English is known as an international language, development of English proficiency and speaking competency should be emphasized [1]. It is essential to master English to increase one's chances of being admitted to a prestigious university and to obtain more employment opportunities in the international marketplace [2]. Speaking English broadens one's world as it enables connections to be forged between people of different countries, cultures and lifestyles. Hence, being able to speak and communicate in English has become the main concern in education.

Speaking is an interactive process that comprises producing, receiving and processing information in the presence of both the speaker and listener to convey feelings, thoughts and opinions [3]. One of the aims of teaching English is to facilitate pupils to speak the target language fluently and accurately in their daily communication, group discussions and classroom presentations. In Malaysia, deficiency in speaking skill has been a main concern in ESL classrooms. Before the implementation of the Common European Framework of Reference (CEFR), the exam-oriented curriculum caused English lessons to focus solely on tested items such as reading comprehension, grammar and essay writing. Limited time

allocation for listening and speaking leads to the negligence of oral language development in the ESL classroom. According to the results of the Cambridge Baseline Study in 2013, pupils' English performance was below the expected level of proficiency and speaking skill was the weakest among the four skills [4]. Some pupils excel in grammar and vocabulary, but they face hindrances when learning English. This is due to the lack of exposure and the cultural background they are brought up with [5]. English is considered as the second language in most families and schools. Pupils tend to speak their mother tongue at home and at school; they seldom have chance to speak English as people in their environment communicate with them using their mother tongue. Most of the pupils are only exposed to English during English lessons in school. In addition to this reason, teacher-centered speaking lessons and pupils' passive learning attitudes also contribute to low speaking competency. In the teacher-centered classroom, pupils passively accept the linguistic knowledge imparted by their teacher without having opportunities to practice their speech [6]. Teachers should alter their traditional teaching methods and conduct more communicative activities that offer chances for pupils to speak English.

The success of speaking English is measured in terms of the pupils' ability to carry out a conversation in English with correct pronunciation, grammar, good use of vocabulary and fluency. The primary obstacles that deter pupils from speaking are the lack of vocabulary as well as some psychological factors, such as anxiety and fear of making errors [7]. To address the issue of low speaking proficiency among primary pupils, educators have revised the curriculum. The curriculum focuses on communicative-based language activities and upholds fun learning in ESL classrooms. It is essential for teachers to develop captivating and interactive speaking activities to motivate pupils to engage in spoken interaction. Student-centered activities with authentic contexts should be planned to gain pupils' interests and motivation to speak in the lessons [8]. Pupils show their willingness to participate when the topics and materials are related to their own lives [9]. Therefore, teachers should select appropriate learning strategies that cater for pupils' needs and learning preferences so that they feel encouraged to speak without any fear of making flaws [10].

One of the ways to get pupils immersed in a speaking lesson is through games. Using games in teaching and learning transforms the traditional method of transmitting knowledge. The incorporation of games in learning triggers pupils to be autonomous learners and enhances their learning in various fields of knowledge [11]. Undeniably, games help to lower pupils' anxieties and create contexts in which pupils can collaborate with peers in teams to use language meaningfully and in a relaxed way. This is supported by [12], who proclaimed that pupils interact with other players and follow the context presented during the games. There are a wide array of communicative games that can be used to teach speaking, and one of them is board games. Through board games, pupils have to take turns to express their ideas according to the instructions given. Some board games imitate real-life circumstances which subconsciously help pupils to develop social skills and increase their flexibility. Several studies were done in the past to examine the effectiveness of the integration of board games into classroom teaching and learning to improve speaking skill. Gaming literature also displays data on the usefulness of board games in language learning. However, there is no systematic literature review that specifies the use of board games to improve English speaking. With this backdrop, this review attempts to present a comprehensive analysis of research studies to synthesize the findings of past studies related to users' perceptions towards the use of board games in speaking, as well as their usefulness in informing future studies. It is vital to look from users' perspectives to investigate their perception towards the use of board games in learning speaking. By understanding this, it helps researchers to evaluate the success of implementing board games in speaking lessons.

Firstly, researchers generally defined board games as tabletop games that involve pieces moved or placed on a board or cards. Board games often include elements such as cards, roles playing and miniature games, according to a set of rules. To determine the appropriateness of the past research studies, which will be reviewed, the researchers

have outlined two research questions: a) What are the pupils' perceptions towards the use of board games to improve their speaking skills? and b) To what extent are board games useful to improve pupils' speaking skills? These questions serve as yardsticks for researchers to select suitable studies to be included in this systematic review. After reviewing the studies, results will be synthesised to answer both questions.

## 2. Literature Review

### 2.1. Speaking Skill

Speaking skill is one of the eminent skills to master in the process of learning English. It is a two-way interactive process which involves producing, receiving and processing information in the presence of both a speaker and a listener to convey feelings, thoughts and opinions [3]. According to [13], speaking is the ability in pronouncing words to express ideas in conversation or dialogue. A fluent speaker is able to pronounce phonemes accurately with the appropriate stresses and intonation in connected speech [14]. Speaking does not only emphasize the articulation of sounds in expressing thoughts, it also looks into fluency and grammar. According to [15], pronunciation refers to the ability to produce speech using correct stress, rhythm, and intonation. Fluency is the ability to deliver speech in a natural colloquial flow without pauses and hesitation. It does not emphasize the flaws that speakers make in terms of grammar, pronunciation and lexicals, as long as the audience comprehends the content [16]. Grammar is described as the correct use of tenses and combination of linguistic units in producing speech [17]. By mastering these elements of speaking, learners are able to convey their ideas successfully.

### 2.2. Board Games

Board games are games that require players to move counters or pieces in particular ways on a pre-marked board, according to a set of rules. Board games can be classified into classic games, family games, strategy games, thematic games and war games. There are several famous board games such as chess, Monopoly, Snakes and Ladders, etc. As well as these traditional board games, there are some strategy board games that involve role playing, for instance, Werewolf, Avalon and Mafia. Players have to use strategy in order to achieve certain goals in these games. Board games are bound by a set of rules and relate to certain contexts. The context relatedness in games is beneficial for children in learning a language as they can share better experiences as they interact with other players during the game. As stated in the theory of social constructivism by [18] in 1978, playing board games helped to enhance children's development [19]. Therefore, board games are recognized as ideal language learning tools as they equip children with new knowledge whilst also entertaining them.

A well-developed board game that matches with specific learning objectives contains high educational value and is highly practicable in a classroom. As most pupils have an interest in games, they will be driven to follow the game rules and speak the target language of the games. They will acquire the language subconsciously as they generate larger vocabularies of language through the games. In foreign countries, board games have been widely used as a technique in teaching speaking skills as board games incorporate elements of turn-taking and cooperative learning which require every player to speak, enabling communication to occur naturally among pupils [20,21].

### 2.3. Board Games in Teaching Speaking

Several studies indicated that adopting board games in speaking lessons influences pupils in their cognitive and affective domain. Through board games, pupils show improvement in the five aspects of speaking skills: grammar, pronunciation, fluency, vocabulary and content [22]. For example, the modified Monopoly and Snakes and Ladders board games act as catalysts that assist pupils to speak in the context of descriptive texts. Board and dice games are also proven to improve pupils' grammar in their speech [23]. The LOSS board game also helps pupils to improve their pronunciation, as they are able to pronounce different words with the same ending sounds.

Moreover, the findings from past studies also reveal that board games engage pupils' participation and improve the speaking ability of low-proficiency pupils in speaking lessons [24,25]. Sometimes pupils cannot express their ideas as they lack the vocabulary. Through board games, pupils are exposed to ideas and opinions given by their peers [26,27]. They listen and learn from their group members' points of view, which helps them to monitor their speaking progress and improve their speaking competence. Advanced pupils acquainted with the games' mechanics can support and assist weak pupils during the games [28,29]. Board games also create a fun and relaxing ambience that lowers pupils' anxieties and motivates them, as they are related to what pupils find interesting and different from regular classroom speaking activities. It is vital to create a cordial and relaxed atmosphere to lower pupils' affective filters and amplify their learning performance [30]. Additionally, some board games connect to real life circumstances which provide opportunities for pupils to share their thoughts and ideas in the target language freely [31]. This assists pupils to be more flexible in the real world and develop their social skills.

Numerous studies have been done by past researchers to explore the usefulness of communicative language activities to improve speaking skills. However, there are very limited resources that focus on pupils' perceptions towards the use of board games in improving their speaking skill and the usefulness of board games to improve speaking skill among pupils. To address this problem, a systematic review was conducted to answer the following questions:

RQ1: What are the pupils' perceptions towards the use of board games to improve their speaking skills?

RQ2: To what extent are board games useful in improving pupils' speaking skills?

## 3. Methodology

In this study, the review and analysis were conducted systematically based on articles and journals from the period, 2017–2021. The articles were retrieved from three databases: Google Scholar, ERIC and SAGE Journals. The review conformed to the PRISMA statement (Preferred Reporting Items for Systematic Reviews and Meta-Analyses), a recognized evidence-based standard which comprises a 27-item checklist and a 4-phase flow diagram. The PRISMA statement guides the researcher to identify inclusion and exclusion criteria in searching related articles and studies as well as to examine large databases of literature reviews in a distinct period.

This review began with the process of collecting findings on the effectiveness of board games in speaking and pupils' perceptions towards the use of board games through the search engines: Google Scholar, ERIC and SAGE Journals. The data collected underwent four phases: identification phase, screening phase, eligibility phase and inclusion phase.

### 3.1. Phase 1: Identification

The data collected for this systematic literature review are from three main sources: Google Scholar, ERIC and SAGE Journals. Google Scholar is a web search engine released in November 2004 that enables users to access freely the full texts or reviews of online academic journals, conference papers, reports, theses and other scholarly literature. in January 2018, Google Scholar was recognized as the largest academic search engine. It is used worldwide and it is estimated to consist of approximately 389 million documents. Meanwhile, ERIC is an online library supported by the Institute of Education Sciences (IES) of the U.S. Department of Education which provides varieties of approved journal and article sources that conform to its standard of selection. SAGE Publishing, founded in 1965 by Sara Miller McCune, is a site that publishes online journals, books and reference works covering fields of humanities, social sciences, science, technology, business and medicine.

The keywords used for the literature search included "board games", "speaking", "ESL classroom" and "students' perceptions". The search identified 128 studies from Google Scholar, 42 studies from ERIC and 55 studies from SAGE Journals. These key words were typed together to narrow the scope and to prevent information that was irrelevant to

the research topic. The studies were then screened according to their year of publication, language, and literature type (Table 1).

**Table 1.** The eligibility and exclusion criteria.

| Criterion | Eligibility | Exclusion |
|---|---|---|
| Years of publication | Between 2017 and 2021 | <2017 |
| Language | English | Non-English |
| Literature type | Journal, research, articles | Non-journals, book chapters, abstracts |

### 3.2. Phase 2: Screening

Cautious screening was done to filter duplicate articles and research papers from the three research engines and the remainders were examined according to the criteria set earlier. Some articles were removed as they did not grant full access for the reviewer. At last, a complete reference list of selected articles was processed.

### 3.3. Phase 3: Eligibility

Several conditions of eligibility and exclusion have been listed to identify qualified articles. Firstly, the articles chosen were published within a period of five years (2017–2021) to ensure the information is relevant and relates to the use of board games in modern education. Next, articles published in other languages were excluded to prevent translation problems and hindrances in understanding. Literature types were taken into consideration and book chapters, abstracts and non-journals are omitted. As the review emphasizes the field of education, articles that related to other areas such as engineering, business, medicine and science were taken out.

### 3.4. Phase 4: Inclusion

At the final stage, only 35 articles that achieved the requirements were enclosed for the review. The articles selected include quantitative research, qualitative research, mixed method research, action research and experimental research published between 2017 and 2021. These criteria were taken into account to produce a high-quality systematic review. The details of the data collection process using the PRISMA flow are summarized in Figure 1.

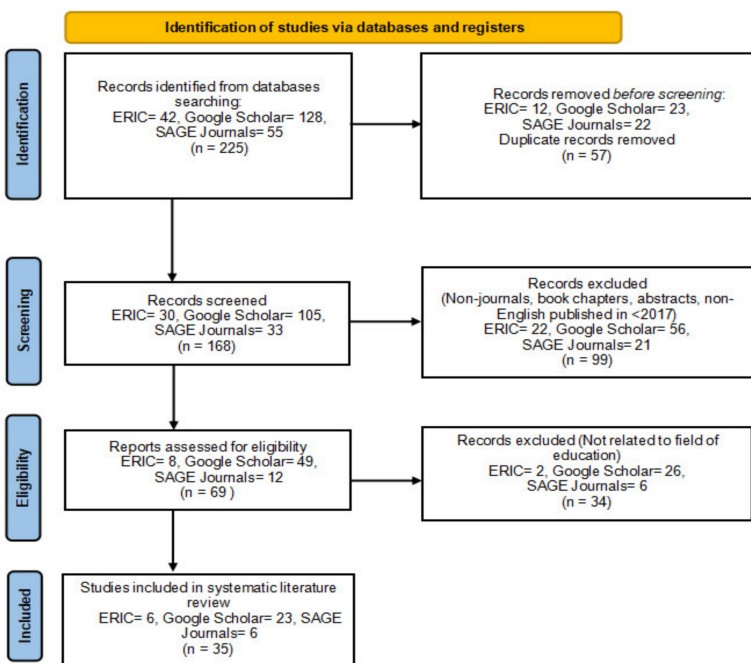

**Figure 1.** Flow chart of the research article selection process.

## 4. Results and Discussion

### 4.1. Pupils' Perceptions to the Use of Board Games

The aggregated results of past studies related to pupils' perceptions of board games are reviewed in this section. Pupils' perceptions to the use of board games were presented in the scope of their learning attitude, learning competence and learning community development. The studies were classified into three categories: quantitative study, qualitative study and mixed-method study. The summary of the list of review studies is shown in Table 2.

**Table 2.** List of reviewed studies of pupils' perceptions to the use of board games.

| Methods | Author (Year) | Attitude | Competence | Community Development |
|---|---|---|---|---|
| Quantitative study, n = 5 | [32] | | / | |
| | [33] | / | | |
| | [23] | / | / | |
| | [34] | / | / | |
| | [35] | | / | |
| Qualitative study, n = 9 | [36] | / | | |
| | [27] | | | / |
| | [29] | | | / |
| | [37] | / | / | |
| | [38] | / | | |
| | [39] | / | | / |
| | [26] | | / | / |
| | [21] | / | | / |
| | [40] | / | | |
| Mixed method, n = 3 | [28] | / | | / |
| | [41] | / | / | |
| | [42] | / | / | |

A collection of 17 articles and journals focused on pupils' perceptions of the use of board games were reviewed. Among them were five quantitative studies, nine qualitative studies and three mixed-method studies. This systematic review explained pupils' perceptions in terms of their learning attitude, learning competence and learning community development. Pupils' learning attitude was recorded as the highest among these three aspects. Thirteen papers reported that pupils agreed that board games increased their motivation in speaking lessons [21,23,28,33,34,36–42]. In eight studies, pupils revealed that their speaking competence improved [23,26,32,34,35,37,41,42]. Lastly, six studies identified that pupils proclaimed that board games enhanced their learning community development [21,26–29,39].

Fithriani [23] identified that pupils themselves reported that board games assisted them in gaining confidence in using English during speaking lessons. Board games lowered their speaking anxiety and provided a game-like atmosphere which enabled them to express their ideas freely within the context provided. Similar findings were depicted in the studies of Łodzikowski & Jekiel [37] and Gonzalo-Iglesia et al. [28]. The former delineated board games as alternatives that promoted active participation in a speaking lessons, while the latter portrayed board games as activating catalysts that motivated pupils to speak in the classroom. For instance, pupils proclaimed that the use of board game, Snakes and Ladders, reduced the boredom of a speaking lesson as it was an engaging and fun game that motivated them to speak confidently in front of the class; they enjoyed many aspects of the game [38]. From pupils' perspectives, board games made speaking lessons fun, reduced their shyness in speaking and lessened their fear of making flaws while speaking [34,42]. Pupils were encouraged to speak as board games replaced drilling practices in speaking lessons [40].

Apart from the change in learning attitude, there was a shift in pupils' speaking competence in terms of grammar, pronunciation and fluency [23,26,27,32,34,35,37,41,42]. Pupils

claimed that board and dice games enabled them to learn how to use tenses, passive voice and conditional sentences in their speech [23]. Pupils admitted that they actively converted the sentences in the board games to their past tense forms and spoke the sentences while they played the game with their peers [42]. Pupils acquired the grammar in the speech subconsciously and improved their speech accuracy gradually through board games [41]. Pupils also perceived board games as a great tool that improved their vocabulary, fluency, grammar and pronunciation in their speech as their scores in the speaking post-test were higher when compared to the pre-test [26,32]. Pupils also viewed board games as helpful in improving their English pronunciation in terms of stress and intonation [37]. Pupils expressed that board games improved their fluency in speaking as they enjoyed playing the games and forget their fear of making errors in their speech [34,35].

Through board games, pupils' learning community development was enhanced in terms of their social ability and communication [21,26–29,39]. Pupils' social skills were improved through board games as they learnt to tolerate losing the games, listened to others' opinions and shared their points of view [27]. Pupils expressed their preferences for board games because they promoted collaborative learning and pupils could socially interact as well as develop teamwork skills with peers [28]. Karasimos' study [29] revealed that pupils showed an active participation in board games as they loved to work with peers in groups.

Based on the above findings, board games are proven to increase motivation in speaking, improve speaking competence and enhance social interaction from pupils' perspectives.

### 4.2. Usefulness of Board Games in Improving Pupils' Speaking Skill

The selected studies have reported the usefulness of board games in improving pupils' speaking skills. The studies are classified into three categories: quantitative study, qualitative study and mixed-method study. Table 3 displays a list of studies related to the usefulness of board games.

**Table 3.** List of reviewed studies on the usefulness of board games in improving pupils' speaking skills.

| Methods | Authors | Exposure | Motivation | Pronunciation | Fluency | Grammar |
|---|---|---|---|---|---|---|
| Action Research, n = 3 | [15] | | | / | / | / |
| | [31] | | | / | / | |
| | [43] | | | / | / | / |
| Experimental, n = 3 | [44] | | | / | | / |
| | [45] | / | | | | |
| | [46] | / | | / | | / |
| Quantitative study, n = 4 | [47] | | | / | / | / |
| | [17] | | / | / | / | / |
| | [48] | | / | / | / | / |
| | [49] | | | / | | / |
| Qualitative study, n = 3 | [50] | | / | / | | |
| | [51] | / | / | | | |
| | [52] | | | / | / | / |
| Mixed-method, n = 3 | [53] | / | | / | | |
| | [7] | / | | / | | |
| | [54] | / | | | / | / |
| Research and development, n = 2 | [55] | | | / | | |
| | [56] | | / | | | / |

Eighteen studies emphasising the usefulness of board games were reviewed. These studies includes three action researches, three experimental researches, four quantitative studies, three qualitative studies, three mixed-method studies and two research and development studies. This systematic review examined the usefulness of board games in improving pupils' speaking in terms of exposure, motivation, pronunciation, fluency and grammar. Pupils' improvement in pronunciation was recorded as the highest among these

five criteria. It was reported in 11 studies that pupils' pronunciation improved after the implementation of board games in speaking lessons [15,31,43,44,47–50,52,54,55]. Furthermore, through board games, pupils were able to speak fluently [15,17,31,43,46–48,52] and make less grammatical errors in their speech [15,17,43,44,47–49,52,54,56]. In addition to this, board games exposed pupils to a myriad of speaking opportunities in different speaking contexts [7,45,46,51,53,54] and motivated pupils to speak [7,17,46,48,50,51,53,56].

Board games were proven to be efficacious in improving pronunciation in terms of clarity [15,31] and intelligibility [43,47]. Pupils were able to identify, blend and segment individual sounds [48], know the differences between phonemes and pronounce the ending sounds /s/, /z/ and /iz/ clearly [55]. Pupils were able to articulate the speech sound with the right pitch [54], stress [50] and intonation [43] with reference to some standard of correctness and acceptability. Repetition of words mentioned by peers in the board games facilitated pupils' memory retention of the words' pronunciations [49].

Board games also caused a significant improvement in fluency. Pupils' speech fluency showed a huge progression as their speed of speech production was maximized [52], grammatical accuracy was increased [46] and they could further elaborate on their points of view [17]. Pupils could understand the conversation and respond to their peers using comprehensible speech [47] as well as express their opinions without hindrances [48]. Halts, repetitions, fillers and sentence fragments were reduced in the speech and communication was not impeded by minor grammatical errors and language limitations.

Apart from the improvement in speech fluency, board games assisted pupils in using grammatical forms such as verb tenses, linking words and conjunctions in their speech correctly [49,52]. Pupils began to use complex and compound sentences in their speech instead of simple sentences [44]. Pupils were aware of the units and patterns used in the speech [48] and developed a relatively high degree of grammatical control [17]. They would select the right tenses and word class while conversing with their peers in board games.

Board games were regarded as tools that exposed pupils to various chances to practice speaking the target language naturally. Pupils were connected to real-life situations that provided meaningful learning opportunities through board games [45,53], thus stimulating them to develop their creativity and thinking into a wider area [46]. Pictures and words provided on the cards in board games enabled pupils to develop ideas for their speech content and pupils gained experience using the language naturally for their daily communication [7].

It was notable that pupils were keen on speaking after board games were implemented in their speaking lessons. A few studies indicated that board games created a positive learning environment with a comfortable atmosphere that reduced apprehension in communication [53] and boosted confidence in speech [56]. Moreover, board games required pupils to gather in small groups to play [7], so pupils were able to learn through their friends' speaking. The stress-free ambience offered by board games allowed pupils to forget their shyness and express ideas naturally [46]. As pupils were familiarised with the concepts, rules and regulations of the board games such as Monopoly and Snakes and Ladders, their anxiety and phobia in speaking was minimized [51] and they felt more encouraged and confident in conveying their opinions to their friends in the games [17].

## 5. Conclusions

To summarize, this systematic literature review analyzed pupils' perceptions on the use of board games and the usefulness of board games in improving pupils' speaking skills. Pupils' perceptions were analyzed in terms of their learning attitude, learning competence and learning community development. The results showed that pupils enjoyed board games as the fun and relaxing gaming atmosphere encouraged them to speak English without feeling inferior or the fear of making mistakes. The nature of board games enabled fun learning and motivated pupils to speak, as they differed from other speaking activities that constrained pupils to learn English according to norms and mundane routines. Addi-

tionally, pupils agreed that board games enhanced their learning community development as they had more social interactions with their peers and learned the viewpoints shared by their peers during the games. The studies highlighted in the systematic review also proved that board games were impressive tools that improved pupils' speaking competence in terms of pronunciation, fluency and grammar. The main speaking obstacles like lack of vocabulary and difficulties in pronunciation decreased. It was discovered that pupils were able to speak with correct pronunciation using correct grammatical forms without many pauses during the games. Out of the 18 studies reviewed on the usefulness of board games in improving pupils' speaking skills, 11 mentioned that board games improved pupils' pronunciation, 8 mentioned that pupils' fluency developed and 10 mentioned pupils were aware of the grammar in their speech. The articles gathered in this systematic literature review answered both research questions in the current study and evidently showed that board games were effective in improving pupils' speaking skills. Therefore, educators should integrate board games into teaching methods to arouse and cultivate pupils' interests to speak English in their daily communication and help pupils to master their speaking skills.

## 6. Limitations and Recommendations

This systematic review offers information to educators in Malaysia on the use of board games in improving pupils' speaking skill. Most articles portrayed that pupils gave positive feedback on the use of board games to learn speaking. The findings could shed light on the implementation of board games to teach speaking in the classroom. There are a few limitations worth discussing in this review that can be further explored by researchers in the future. Firstly, this review only managed to analyze 35 articles from ERIC, Google Scholar and SAGE journals. Secondly, the review revolves around articles that report students' perceptions on the use of board games. It is recommended that future researchers explore more databases to obtain findings from different sources to enrich and expand the study. Hopefully, future studies can look into both teachers' and students' perceptions on the usefulness of board games in teaching and speaking so that other educators can learn opinions from both perspectives.

**Author Contributions:** All authors contributed to several aspects of the study, specifically, conceptualization, C.H.T.W. and M.M.Y.; methodology, M.M.Y.; validation, M.M.Y.; formal analysis, C.H.T.W.; investigation, C.H.T.W.; resources, M.M.Y.; data curation, C.H.T.W. and M.M.Y.; writing— original draft preparation, C.H.T.W.; writing—review and editing, C.H.T.W. and M.M.Y.; supervision, M.M.Y.; project administration, C.H.T.W.; funding acquisition, M.M.Y. Both authors have read and agreed to the published version of the manuscript.

**Funding:** This research was funded by Universiti Kebangsaan Malaysia under research grant number GG-2019-006 and the APC was funded by Universiti Kebangsaan Malaysia.

**Institutional Review Board Statement:** Not applicable.

**Informed Consent Statement:** Informed consent was obtained from all subjects involved in this study.

**Data Availability Statement:** Not applicable.

**Conflicts of Interest:** The authors declare no conflict of interest.

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
