# Peer review of "Board Games in Improving Pupils’ Speaking Skills: A Systematic Review"

_sustainability, doi:10.3390/su13168772_

Round 1
Reviewer 1 Report
Although this paper has some positive aspects, several features need to be addressed before performing a more in-depth review:
1. Theoretical framework: Although there is a short definition of ‘board games’, no attempt has been made to illustrate or classify them in relation to speaking or depending on the 17 articles reviewed. This is a very broad concept which requires further clarification. Similarly, ‘speaking’ is also a broad concept as it may refer to fluency, accuracy, pronunciation, etc. However, these concepts are not clearly defined in the paper.
2.Although the paper is said to focus on the use of ‘board games’ to improve the ‘speaking’ competence, this competence needs to be better defined. ‘Learning attitudes’ and ‘learning community development’ may have an impact on speaking (reduced language anxiety, etc) but the authors could have examined other more linguistic factors such as fluency, pronunciation, etc. (see Table 4 - paper). In fact, they refer to some of these factors such as ‘pronunciation, intelligibility, etc’ as analysed in the 17 articles reviewed, but these linguistic factors are not the core subject of the paper.
3. OBJECTIVES: The second objective needs to be reworded, the agrammatical nature of the sentence leaves its meaning unclear: ‘To what extent is the usefulness of the board games in improving pupils’ speaking skill?’ reconsider ‘to what extent … usefulness’ (92-93 & 144) (for example: To what extent are board games useful to improve pupils’ speaking skills? Or What is the usefulness of board games in …. ?)
4. Some conclusions are not fully proven, for example the statement ‘To summarize, pupils’ upbringing background and the negligence of the importance of speaking skill in the language learning process lead to their poor speaking performance’ (312-313). There is no clear reference to these variables in the articles reviewed in this paper, only some general allusions to the national context of the author/s of this review, Malaysia. However, the 17 articles analysed may have been produced in rather different contexts and countries where the educational system could be quite different, therefore the authors should not draw conclusions based on their local context, not on the articles analysed. There seems to be no clear correlation between this statement and the results of the systematic review performed.
5. The conclusion section is quite short and weak, just one paragraph with general ideas, and the statements included are not directly related with the factors previous analysed in the systematic review or the 17 articles. The authors are recommended to concentrate on the results of their systematic review rather than on their experience in their local context (Malaysia) since the 17 papers reviewed may have not been presumably written in the same context.
English language editing is needed, some examples are provided below:
(Lines 24-26) ‘English is used as a means of verbal and written communication worldwide to bridge the gap in economical, political and social aspects’ (economic?)
(Lines 60-61) ‘It is essential for teachers to develop engaging and interactive speaking activities to motivate pupils to engage in spoken interaction’ (avoid repeating engage, engaging)
I feel that these issues need to be addressed as a primary step to undergo a more thorough in-depth revision.
Author Response
Dear Reviewer,
Thank you for your reviews. Changes have been made based on the reviews provided. The point-by-point response is written in the document as attached. Please refer to the attachment. Thank you. Have a nice day.
Best regards.

Reviewer 2 Report
Dear authors,
interesting topic.
Please rewrite the abstract as it has repetitions and even mistakes:
- Previous researchers explored a myriad of communicative language activities to improve speaking skills. Nevertheless, the researches on the usefulness of board games to improve speaking among pupils and pupils’ perceptions towards the use of board games have been very limited.
Pay attention to language:
- The studies were then screened according to its their year of publication, language, and literature type (Ta- 175
ble 1).
Also the RQ2 does not sound natural: To what extent is the usefulness of board games in improving pupils’ speaking skill?
maybe to what extent are the board games useful ....
What about studies in your own countries? have you found any?
Author Response

(The authors gave the same response as above.)

Reviewer 3 Report
Board Games in Improving Pupils’ Speaking Skill: A Systematic Review 3
Catherine Hui Tiing Wong and Melor Md Yunus *
… speaking English has been highlighted among Malaysian pupils as English plays a major role in 8 global communication…
SPEAKING OR ALL SKILLS?
14 through three databases namely Google Scholar, ERIC and SAGE Journals.
Why were these chosen, any other relevant or consulted and was not selected for the study?
28 It is essential to master English to increase the chances to be admitted to a prestigious university and to obtain more employment opportunities in the international marketplace.
Agree it broadens chances in life, but not all do it to get into prestigious universities, source?
37 In Malaysia, incompetence in speaking skill
Would change the word “incompetence” to something softer
38 Common European 38 Framework of Reference (CEFR), the exam-oriented curriculum
The CEFR is not exam oriented, it´s a language framework to facilitate planning, levelling, materials….
46 English is considered 46 as the second language in most families. Pupils tend to speak their mother tongue at home, 47 most of them are only exposed to English during the English lessons in school
Why families? Maybe contexts, schools,… that´s why you are dealing with Foreign Language, as the surrounding context does not use English.
52 Teachers should reflect on their teaching methods and 52 devise activities that offer chances for pupils to practice their speaking.
I think in this paragraph many core concepts and terms are being mentioned but don´t seem to be very clear, maybe simplify and make it more coherent
80 there is no systematic literature review to analyse the results
Gaming literature shows data and studies about board games
104 roles playing
Typo: Role playing
109 Lev Vygotsky (reference, mentioning)
116 In foreign countries, board games have been widely used as 116 a technique in teaching speaking skill
Foreign?
123 The modified Monopoly and Snake and Ladder board games
For example: the modified…..
125 reveal that board games engage pupils’ participation and improve the speaking ability of low proficiency pupils…
And high proficiency pupils?
127 By using board games, they can follow the instructions given and know what sentence to say…
Due to? Not because they are given the instructions, they can follow them, but maybe because there was a pre-activity explaining the game, or visual reinforcement and/or motivation or teamwork….
146 Information from electronic media and printed resources could be exploited when doing a literature review
Non relevant
147 the review and analysis were conducted systematically from March 2021 to June 2021
When the dates are mentioned, do they mean the time spam used to do a review or the articles published between those dates? In line 184 some more info about the time frame is given.
172 and on The keywords used for the literature search include “board games”, “speaking”, 172 “ESL classroom” and “students’ perceptions”. The search identified 128 studies from 173 Google Scholar, 42 studies from ERIC and 55 studies from SAGE Journals. The studies 174 were then screened according to its year of publication, language, and literature type (Ta-175 ble 1).
Seems to me that if you search based on your key words “ESL classroom””Speaking”…. Many many studies will be shown, or were all words together in the search. Maybe rephrase for clarification.
185 and follows the latest trend of education
Using game boards? Rephrase
187 Literature types are taken into consideration in which book chapters, abstracts and non-journals are omitted.
Please specify why and under which basis
201 4. Results and Discussion
Please clarify if results and discussion takes into account the information retrieved from the papers selected or takes into account any other studies. And if results consider any foreseeable practices, recommendations or opinions.
213 Among them are 5 quantitative studies, 9 qualitative 213 studies and 3 mixed method studies
Non-journals and book chapters were not taken into consideration, maybe this kind of publications show results of studies of different nature.
256 The studies are classified into three categories: quantitative study, 256 qualitative study and mixed method study. Table 3 will display a list of studies related to 257 the usefulness of board games.
Any relevant data proving if results will be different whether gathered from a qualitative, quantitative or mixed method?
261 18 studies emphasising on the usefulness of board games have been reviewed.
In the introduction the author mentions that “A total of 35 articles were accessed”
327 This systematic review offers information to educators in Malaysia on the use of board games in improving pupils’ speaking skill.
Was the material reviewed of a specific geographical context?
332 due to restriction of time, this review only manages to analyse 35 articles from ERIC, Google Scholar and SAGE journals
Explain the restriction of time, is this review part of a bigger research project?
Reading the title is an attractive article, but while reading I get the feeling that the title does not really reflect on the core topic of the article. I felt a bit lost at times, as whether the study shows a literature review as far as format of presentation of studies (way of referencing, mentioning, key words, types of research) or an article about the use of board games in the FL class.
It needs to be clearer, some paragraphs are not relevant and some others need more explanation as to be able to really get the idea the authors want to transmit.
Author Response

(The authors gave the same response as above.)

Round 2
Reviewer 3 Report
The article seems to have really improved, thanks to the authors for making the suggested changes.